# Identification of Novel Fusion Transcripts in High Grade Serous Ovarian Cancer

**DOI:** 10.3390/ijms22094791

**Published:** 2021-04-30

**Authors:** Andreea Newtson, Henry Reyes, Eric J. Devor, Michael J. Goodheart, Jesus Gonzalez Bosquet

**Affiliations:** 1Department of Obstetrics and Gynecology, Division of Gynecologic Oncology, University of Iowa Hospitals and Clinics, Iowa City, IA 52242, USA; michael-goodheart@uiowa.edu (M.J.G.); jesus-gonzalezbosquet@uiowa.edu (J.G.B.); 2Department of Obstetrics and Gynecology, University of Buffalo, Buffalo, NY 14260, USA; henrypogs@yahoo.com; 3Holden Comprehensive Cancer Center, University of Iowa Hospitals and Clinics, Iowa City, IA 52242, USA; eric-devor@uiowa.edu; 4Department of Obstetrics and Gynecology, University of Iowa Hospitals and Clinics, Iowa City, IA 52242, USA

**Keywords:** fusion genes, fusion transcripts, high grade serous ovarian cancer, whole transcriptome sequencing

## Abstract

Fusion genes are structural chromosomal rearrangements resulting in the exchange of DNA sequences between genes. This results in the formation of a new combined gene. They have been implicated in carcinogenesis in a number of different cancers, though they have been understudied in high grade serous ovarian cancer. This study used high throughput tools to compare the transcriptome of high grade serous ovarian cancer and normal fallopian tubes in the interest of identifying unique fusion transcripts within each group. Indeed, we found that there were significantly more fusion transcripts in the cancer samples relative to the normal fallopian tubes. Following this, the role of fusion transcripts in chemo-response and overall survival was investigated. This led to the identification of fusion transcripts significantly associated with overall survival. Validation was performed with different analytical platforms and different algorithms to find fusion transcripts.

## 1. Introduction

Though much effort has been invested in defining the biology and natural history of ovarian cancer, standard treatment and prognosis has not changed much since the addition of platinum based chemotherapy [1]. A notable exception is a subset of ovarian cancer patients with deficiencies in DNA repair who have enjoyed new targeted therapies with significant gains in progression free survival [2,3,4,5]. However, for as many as 50% of patients, the biology of their disease is not completely clear and has not yet been targetable [6]. The Cancer Genome Atlas (TCGA) made great strides in defining the genomic profile of high grade serous ovarian cancers. However, beyond and/or because of a high preponderance of TP53 mutations and homologous recombination deficiency, the genomic profile was marked by disarray, especially in contrast with other cancer types, such as glioblastoma multiforme [6]. As such, it is necessary to expand the repertoire of examined genomic features in high grade serous ovarian cancer. Fusion genes represent an understudied genomic feature of high grade serous ovarian cancer. 

A fusion gene is a structural chromosomal rearrangement resulting in the exchange of DNA sequences between genes. This results in the formation of a new combined gene. Fusion genes have been implicated in carcinogenesis since the early 1980s. At that time, banding techniques allowed for chromosomal analysis of tumors, which led to the discovery of *BCR-ABL1* as a recurrent structural chromosomal rearrangement implicated in chronic myeloid leukemia (CML) [7]. This then allowed for an exceptionally successful targeted therapy, Imatanib. Until recently, fusion gene identification was biased towards interchromosomal rearrangements due to (1) the difficulty of culturing cells at a specific phase in the cell cycle and (2) differentiating genomic “noise” from pathogenetically important aberrations. However, modern high-throughput tools allow investigators to perform genomic analyses with enough granularity to identify significantly more fusion genes in cancers, including solid tumors. Deep sequencing technology permits even greater granularity, such that subtle intrachromosomal rearrangements can now be identified [7], and constantly advancing bioinformatic techniques aid investigators in differentiating “noise” from pathogenic rearrangements [8]. Examples of fusion genes discovered with the help of modern genomics include: the *TMPRSS2-ERG* in prostate cancer [9], *RET-CCDC6* in thyroid carcinoma [10], and *EML4-ALK* in non-small cell lung cancer (NSCLC) [11]. Remarkably, fusion genes such as *NTRK* fusion genes have also been identified as driver mutations in a variety of different adult and pediatric cancers, and “tumor agnostic” therapies targeting these fusion genes have resulted in impressive treatment responses in phase I and II clinical trials [12]. 

Similar work has started with ovarian cancer. Indeed, genetic rearrangement has been identified as a mechanism of tumor suppressor inactivation in ovarian cancer. Genomic rearrangement may result in the formation of fusion genes [13]. Investigators have recently begun utilizing deep sequencing to improve upon previous work done using guided techniques [13,14,15]. This work has led to the identification of gene fusions in *NRG1* and *ABCB1*, the latter of which was specifically identified in pre-treated and drug-resistant specimens. This work suggests that fusion genes do play some role in the pathogenesis high grade serous ovarian cancer, recurrence, and/or adaptive drug resistance [15,16]. Because the most significant morbidity of ovarian cancer lies in drug-resistant recurrences following heavy pre-treatment, analyzing tumor genomics in the context of clinical outcomes, such as survival and chemo-response is crucial. As such, most ovarian cancer fusion gene work has focused on this subset of poor-prognosis, heavily pre-treated, and chemo-resistant ovarian cancer patients, but some clues may lie within treatment-naïve primary presentations. Indeed, there may even be some clues within pathologically normal fallopian tubes themselves.

Our hypothesis is that fusion genes are part of the genomic rearrangements which occur in ovarian cancer. This study aims to assess differences between fusion transcripts’ presence in primary high-grade serous ovarian cancer (HGSC) and normal fallopian tube samples. We then determine whether these fusion transcripts are associated with clinical outcomes, specifically survival and response to chemotherapy. In this manuscript, “fusion transcript” refers to the sum formation of two partner transcripts.

## 2. Results

### 2.1. Fusion Transcript Differences between Fallopian Tube and HGSC Samples

A total of 597 fusion transcripts were identified within all samples (Figure 1). Appendix A details all fusion transcripts observed in tubal and HGSC samples, as well as their position in chromosomal references. We found more fusion transcripts in HGSC samples (Figure 1E). On average, there were 6.59 fusion transcripts present in each cancer sample and 3.08 in tubal samples (chi-square *p* < 0.001). Figure 1A shows both components of the fused transcript in a circular chromosomal representation. In the univariate analysis, there were 3 fusion transcripts with significant different frequencies between fallopian tube and HGSC samples: *AL391840.3—SH3BGRL2*, *AL445985.1—SPATA13*, and *PFKFB3—LNC02649* (Figure 1B). These were all intrachromosomal rearrangements, and the fusion transcripts in the normal fallopian tubes were significantly shorter than those found in HGSC samples. Appendix A. *AL391840.3—SH3BGRL2* is found on chromosome 6, *AL445985.1—SPATA13* is found on chromosome 13, and *PFKFB3—LNC02649* is found on chromosome 10. Notably, all these fusion transcripts were found *less* frequently in HGSC samples than in tubes (Figure 1C). In a multivariate analysis, *AL391840.3—SH3BGRL2* was the only fusion transcript independently associated with HGSC (Figure 1D). 

### 2.2. Prediction Model of HGSC Using Fusion Transcript Data

The fusion transcript HGSC prediction model included 5 different fusion transcripts. The performance of that model, measured in AUC, was 95%, with a 95% CI of 92%, 98% (Figure 2). All identified fusion transcripts in the model were decreased in cancer samples relative to normal tube (Figure 2C). 

### 2.3. Association of Fusion Transcripts with Overall Survival

In the univariate analysis of baseline clinical variables, Charlson Comorbidity Index, disease in the upper abdomen diagnosed by imaging studies, and treatment with neoadjuvant chemotherapy were significantly associated with survival (Table 1). In the multivariate analysis, neoadjuvant chemotherapy was the only variable independently associated with survival (*p*-value = 0.015, Figure 3A). 

In the univariate analysis of fusion transcripts associated with survival, 44 fusion transcripts found within the ovarian cancer cohort were found to be statistically significant at *p*-value < 0.05 (Appendix A, Appendix B). Of the significant fusion transcripts in the univariate analysis, 10 remained independently significant in the survival multivariate analysis with Cox proportional Hazard ratio (Figure 3B, Appendix B).

In the integrative multivariate model that combined independently significant clinical and fusion transcripts variables, all variables remained significantly associated with survival (Figure 3C). Of the fusion transcripts found to be significantly associated with overall survival, the direction of association was towards worse overall survival. Notably, there were two fusion transcripts with a huge association: *ZBTB8OS—AC090627.1* was found to be over 1000 times more associated with worse survival, and fusion transcripts *ARL17A—KANSL1* was over 400 times more associated with worse survival. *ZBTB8OS—AC090627.1* is an inter-chromosomal fusion transcript between chromosomes 1 and 17, and *ARL17A—KANSL1* is an intra-chromosomal fusion transcript on chromosome 17. 

### 2.4. Association of Fusion Transcripts with Chemo-Response

Clinical variables associated with chemo-response included: age, Charlson Comorbidity index, residual disease after surgery (optimal versus suboptimal debulking), and receipt of neoadjuvant chemotherapy. In the multivariate analysis, three variables were found to be independently significant: age, residual disease after surgery, and receipt of neoadjuvant chemotherapy (Figure 4A,B). No fusion transcript was significantly associated with chemo-response. The number of fusion transcripts was not significantly different between responders and non-responders (Figure 4C). 

### 2.5. Validation of Fusion Transcript Detection with FusionCatcher, DNA Sequencing

As a negative control for the method of detection, we extracted DNA from normal tubes, performed DNA sequencing, and applied the *STAR-Fusion* method to the *fastq* files. As expected, we did not find any fusion gene on any of the normal samples.

To validate the fusion transcript detection, we repeated the analysis with *FusionCatcher*, a different method with a different algorithm, to detect fusion transcripts and chimeras. Out of the 44 fusion transcripts found to be significant in the univariate analysis (see *Association of fusion transcripts with overall survival* section), we detected 31 fusion genes with *FusionCatcher* with accuracy detections over 95% and AUC average of 97% (Table 2). Four out of the 10 fusion transcripts which remained independently significant in the multivariate analysis of survival were also accurately detected (AUC of 89%). A new multivariate analysis of survival was performed with these 4 validated fusion transcripts using *FusionCatcher*. All were significant in the same direction of the initial analysis (in the direction of worse survival). Two of the transcripts were significant in the validation of the multivariate analysis of survival (*p* < 0.05): *FAM98B--FRMD5* and *NRIP1--AJ009632.2*. The other two transcripts were close to significance, *ARL17A--KANSL1* and *CC2D1A--CPNE8*, *p* = 0.054 and *p* = 0.074, respectively (Appendix A). 

### 2.6. RT-PCR Validation

Four fusion transcripts identified using *STAR-Fusion* and *FusionInspector* were all confirmed to be present when RT-PCR was performed on the original samples (Figure 5). We chose to validate these fusion transcripts with RT-PCR because we wanted to include some which were validated with the independent analytical platform, *FusionCatcher* (*FAM98B*--*FRMD5*, *CC2D1A*--*CPNE8*), and other fusion transcripts which were not (*AUTS2*---*INO80C* and *AC004475.1*--*PRPF6)*.

## 3. Discussion

The TCGA allowed investigators to examine the genomic underpinnings of ovarian cancer, and such analyses have helped identify a molecular subset of patients (HRD) which benefit from targeted therapy (PARP inhibitors). However, this only represents about 50% of patients. For the other 50% of patients, tumor susceptibilities and prognostic biomarkers are still undiscovered. Fusion genes represent one stone which has been left underexplored in the study of ovarian cancer genomics. 

Fusion genes have provided clarity in the pathogenesis, prognosis, and treatment of other cancers, and technological advances in deep sequencing have allowed researchers to discover the presence of pathogenic fusion genes within a variety of carcinomas—prostate, lung, sarcoma, and even targetable tumor agnostic driver mutations. In ovarian cancer, fusion genes have been linked to drug resistance [16], potentially targetable mutations [15], and associations with rare ovarian tumor types [17]. We used similar techniques to try to identify fusion transcripts within HGSC using whole transcriptome sequencing.

We first sequenced the transcriptome of normal fallopian tubes and primary HGSC. We identified the fusion transcripts present within both tissue types using *STAR-Fusion* and validated these fusion transcripts with *FusionInspector*. We found that three fusion transcripts were present more frequently in normal fallopian tubes than in HGSC. Possible explanations for these “normal” fusion transcripts are (1) the fusion transcripts are protective against carcinogenesis or (2) the “normal” fusion transcripts are a marker of relative genomic stability compared with HGSC. In terms of functional analysis, the gene *SH3BGRL2* has been hypothesized to be a tumor suppressor in renal cell carcinoma [18] and serve a role in the migration and invasion of breast cancer [19]. The function of *AL391840.3* is not yet known. These fusion transcripts were not found in the analysis of fallopian tube DNA, indicating that these findings represent post-transcriptional modifications. Further analyses are needed to evaluate whether there is a functional meaning to these fusion transcripts, or if they are germline level events rather than occurring at the somatic level.

We then used these fusion transcripts to construct a prediction model which aims to identify a fusion transcript’s origin—normal fallopian tube versus HGSC. Such a prediction model would potentially be useful for testing serum cell free genomic testing as a non-invasive diagnostic test of ovarian cancer, or so called “liquid biopsy.” The model performs relatively well with an AUC of 95%. Though, such a model would be difficult to apply in the clinical setting because it relies on the *absence* of fusion transcripts to differentiate the carcinoma from the normal tissue, which may lack specificity. Lack of specificity, indeed, plagues many non-surgical ovarian cancer diagnostics. 

We did not identify any specific fusion transcripts associated with chemo-response, suggesting that they may not represent driver mutations which determine response to treatment. This contrasts with Christie et al.’s work [16], though their analysis was conducted in a heavily pre-treated population, and our samples were primarily obtained from chemotherapy naïve patients (with the exception of those who received neoadjuvant chemotherapy). Thus, our study of drug resistance mechanisms is limited by our study population. A significant number of fusion transcripts were found within our non-responders, suggesting that they may be involved in chemoresistance some way, which would support the findings in Christie et al. Though, a significant number of fusion transcripts were also found within our chemo-responsive samples, indicating that fusion transcripts are diffusely present within ovarian cancer. Patch et al. found that genomic rearrangement and breakage are often found in chemo-resistant ovarian cancers, and fusion transcripts are one such consequence of these rearrangement and breakage events. However, we did not identify the specific fusion events Patch et al. identified as implicated in chemoresistance—those involving *ABCB1*. Though, in contrast with Patch et al.’s work, our samples were obtained from surgeries as part of primary treatment, rather than recurrent disease [13]. Perhaps we could have identified fusions involving *ABCB1* had we analyzed tumor samples obtained at the time of recurrence, rather than primarily evaluating the original tumor. It may be that these fusion transcripts are selected for over the course of multiple treatment lines.

Ten fusion transcripts were associated with significantly worse survival. Two, in particular, had striking hazard ratios for worse survival-- *ZBTB8OS—AC090627.1* and *ARL17A—KANSL1*. Given that these fusion transcripts exist within a large group of fusion transcripts (over 300), it is possible that these were “passenger” mutations which confer some particular survival advantage to the cancer, resulting in a poor prognosis for the patient. These two fusion transcripts may therefore be markers of a particularly aggressive iteration of ovarian cancer. From a functional perspective, *ZBTB80S* is a component of the tRNA splicing complex required to facilitate the enzymatic turnover of the catalytic subunit for RNA-splicing ligase [20]. Though the exact function of *AC090627.1* is not known, it has been identified as a partner fusion gene within breast cancer [8]. It encodes a long non-coding RNA [21]. The *ARL17A—KANSL1* fusion transcript has been identified in both normal and thymus and T cell lymphoblastic lymphoma tumor samples and is thought to be involved in tumor maintenance, rather than pathogenesis [22]. Indeed, Zhou and colleagues identified this specific fusion gene within multiple different cancer types among patients of European ancestry origin, suggesting that this fusion gene may represent a genetic predisposition for cancer within this patient group. Previous study of the *ARL17A—KANSL1* fusion shows some loss of functional domains, and since the unfused version of these genes are involved with histone acetyltransferase KAT8 and p53, the fused versions are hypothesized to interfere with these functions. This may represent a fusion gene which predisposes patients to carcinogenesis, analogous to mutations in BRCA [23]. By itself, the gene *ARL17A* encodes a GTP-binding protein which is suspected to be involved in protein trafficking and may modulate vesicle budding and uncoating within the Golgi apparatus [24]. *KANSL1* is involved with histone acetylation [25]. The two fusion transcripts found to be significantly associated with overall survival using *FusionCatcher* were *FAM98B--FRMD5* and *NRIP1--AJ009632.2*. *FAM98B* is a protein coding gene which counts tRNA processing and gene expression as its related pathways [26], and it has been associated with colon cancer progression [27]. *FRMD5* is also a protein coding gene which is involved in cell migration and has been linked to lung cancer progression [28,29]. *NRIP1* encodes a protein which is involved in transcriptional activation by steroid receptors, including the estrogen receptor [26], which plays a significant role in the development of normal fallopian tubes. Mutations in *NRIP1* have been linked to breast cancer in genome wide association studies [30,31]. *AJ009632.2* is a long non-coding RNA whose function is not yet known, but a variant of this long non-coding RNA has been linked to Parkinson’s disease in a genome wide association study [32]. 

We validated our fusion detection analysis (*STAR-Fusion*) with an independent method (*FusionCatcher*) and found that most of significant transcripts associated with survival were also detected with the new method. Some of those poor prognosis transcripts found to be significant in the *Star-Fusion* analysis trended towards poor prognosis in the *FusionCatcher* analysis. They remained either significant or close to significant in the repeat multivariate analysis. Then, we selected four fusion transcripts from the multivariate overall survival analysis which were found using *STAR-Fusion* and *FusionInspector*: two were identified using *FusionCatcher* and two were identified using *Star-Fusion*. RT-PCR was then performed on these four fusion transcripts, and all were identified within the original patient samples, confirming their true presence within the original transcriptome (rather than an artifact). The validation analysis underscores the complexity of fusion transcript detection and analysis. Discovery of fusion transcripts through computational tools is a developing technology that is evolving constantly [33,34,35]. There is a need for continued development of and standardization of fusion detection tools, candidate fusion prioritization algorithms, and dedicated fusion databases to improve detection accuracy and sensitivity [35]. Moreover, mechanistic analysis are needed to evaluate whether there is a functional meaning from these fusion transcripts, if they reflect the genetic instability of tumors with more aggressive phenotype, or if they are germline-level events rather than occurring just at the somatic level. Continued investigation may also reveal that these or other transcripts provide not just prognostic information but also targetable events, as they are with *BCR-ABL1* [7] and *NTRK* fusions [12].

This study is limited by its retrospective nature and small sample size of normal fallopian tubes. Though the fusion gene software is validated, many others exist. Indeed, we did not identify previously described recurrent fusion transcripts in high grade serous ovarian cancer, such as *ESRRA-TEX40* [36] or *CDKN2D-WDFY2* [37], which could be due to the slightly different ways by which each study identified its fusion transcripts. Thus, as we advance our methods of analyzing next generation sequencing data, it is imperative that we continue looking to old samples to gain new insights.

To further explore these results, we would need a mechanistic approach to identify the actual interaction between fusion transcripts and molecular components of the cellular machinery. This would help define the genomic and cellular mechanisms by which ovarian cancer shortens patients’ lives and doing so may reveal targetable opportunities for intervention—before the patients present with recurrent malignant bowel obstructions, pleural effusions, and the various other clinical conditions by which our patients ultimately succumb to their disease. Indeed, there is no one singular way by which ovarian cancer leads to mortality, so there are likely a multitude of cellular processes to discover, some of which we suggest originates from fusion events. Moreover, though we are presuming that these fusion events are produced at the transcriptional level, it is possible that there are events at the DNA level that occur due to the genomic instability of cancer cells [38]. The combination of both—alteration of tumoral DNA and transcriptional—will give us a more complete picture of the actual fusion events. Differentiating between the two would require both germline and somatic transcriptome analysis of ovarian cancer patients.

Ultimately, fusion genes represent a relatively underexplored genomic feature of ovarian cancer. Deep sequencing and unguided techniques have resulted in fruitful discoveries within other cancer types, which has better informed prognosis and resulted in targeted therapy. As the HRD story shows us, improved understanding of tumor biology and pathogenic subsets of cancer can yield significant clinical advances for both targeted therapy and prognosis. The high preponderance of fusion transcripts in high grade serous ovarian cancer relative to normal fallopian tube tissue makes it clear that fusion genes have some role in carcinogenesis. With the discovery of targetable driver fusion genes such as those involving *NTRK* leading to effective and well tolerated “tumor agnostic” therapies [12], the potential for new treatment opportunities is great. Indeed, fusion genes may represent the next frontier for ovarian cancer. 

## 4. Materials and Methods

This is a retrospective case-control study that used clinical and genomic information to identify fusion transcripts in primary HGSC and normal fallopian tube samples. To assess clinical outcomes, we classified HGSC patients as responders or non-responders to chemotherapy. Responders were those with a progression-free survival of at least 6 months after the first platinum-based treatment. Non-responders were those who had evidence of disease within 6 months of their platinum-based treatment (platinum-resistant) or experienced disease progression during treatment (platinum-refractory). Overall survival was defined as the time from treatment completion to death. Patients who were alive at the end of their follow-up were treated as censored observations.

### 4.1. Patient Inclusion Criteria

We identified ovarian cancer patients with high grade serous histology from available flash-frozen tumor tissues stored at the University of Iowa Hospitals and Clinics Department of Obstetrics and Gynecology Gynecologic Oncology Bank (IRB, ID#200209010) that is part of the Women’s Health Tissue Repository (IRB, ID#200910784). Ovarian cancer patients with clinical and pathological data were included. Patients without RNA of sufficient quality (see below) for RNA-sequencing (RNA-seq) analysis were excluded from the study. Of the 187 patients identified in the original HGSC panel, 112 tumor tissues with sufficient RNA yield and quality were available for analysis.

We additionally analyzed 12 fallopian tube samples from women undergoing salpingectomies for contraceptive indications or as part of a hysterectomy for chronic pelvic pain, pelvic organ prolapse, abnormal uterine bleeding, fibroids, or dermoid cyst. The analyzed tissue came from the junction of the ampullary and fimbriated end of fallopian tubes. For the benign fallopian tubes, a separate approval was given by the University of Iowa Institutional Review Board (IRB, ID#201202714) in coordination with the University of Iowa Tissue Procurement Core Facility. All tissues were obtained from adult patients under informed consent in accordance with the University of Iowa IRB guidelines. 

### 4.2. Clinical Data

Clinical and pathological data were collected from the medical record with IRB approval from the University of Iowa (UI) (IRB ID# 201804817). 

Clinical variables previously observed to be associated with chemo-response were included in the data collection [39]. Only baseline clinical and pathological characteristics which can be obtained before starting initial chemotherapy were included. Differences between clinical variables in responders versus non-responders were assessed by logistic regression. *p*-values ≤ 0.05 were considered statistically significant. There were 103 HGSC patient with accompanying clinical data included in the survival analysis (Table 1); 88 of those patients have complete information about response to treatment and were included in the chemo-response analysis (Appendix A).

### 4.3. RNA Purification and Whole Transcriptome Sequencing

Total cellular RNA was purified from primary tumor tissue using the mirVana (Thermo Fisher, Waltham, MA, USA) RNA purification kit following the manufacturers’ instructions. Yield and quality of purified cellular RNA was assessed using a Trinean DropSense 16 spectrophotometer and an Agilent Model 2100 bioanalyzer. Samples with an RNA integrity number (RIN) [40] greater than or equal to 7.0 were selected for RNA sequencing. Genomic DNAs from frozen normal tubal tissue were purified using the DNeasy Blood and Tissue Kit according to manufacturer’s (QIAGEN GmbH, Hilden, Germany) recommendations. 

Equal mass total RNA (500 ng) from each qualifying tumor was fragmented, converted to cDNA, and ligated to bar-coded sequencing adaptors using Illumina TriSeq stranded total RNA and DNA library preparation (Illumina, San Diego, CA, USA). Molar concentrations of the indexed libraries were confirmed on the Agilent Model 2100 bioanalyzer, and libraries were then combined into equimolar pools for sequencing. The concentration of the pools was confirmed using the Illumina Library Quantification Kit (KAPA Biosystems, Wilmington, MA, USA). Sequencing was then carried out on the Illumina HiSeq 4000 genome sequencing platform using 150 bp paired-end Sequencing By Synthesis (SBS) chemistry. All library preparation and sequencing were performed in the Genome Facility of the University of Iowa Institute of Human Genetics (IIHG).

For quality control of our RNA-Seq experiments, we looked at the number of reads per sample and number of unmapped transcripts (average read/sample over 27 million). Unmapped transcripts resulted from: (1) transcripts being too short for successful mapping (less than 200 bp) and represented an average 21% of all transcripts; (2) 0% of transcripts had too many mismatches, and (3) 0.2% for other causes. None of the samples had a number of mapped reads below the 10-million threshold.

### 4.4. DNA Extraction from Normal Fallopian Tubes

Genomic DNA from fresh frozen normal fallopian tubes were purified using the DNeasy Blood and Tissue Kit according to manufacturer recommendations. Yield and purity were assessed using a NanoDrop Model 2000 spectrophotometer with a 260 nm/280 nm absorbance ration of ~1.8 with minimal to no degradation seen using horizontal gel electrophoresis. Following purification, the samples were bisulfite-converted using the EZ-96 Deep-Well Format DNA Methylation Kit following the Illumina Infinium^®^ Methylation Assay alternate incubation instructions. Of the 20 collected fallopian tubes, 12 met the quality standards.

### 4.5. Fusion Transcript Detection

We used the *STAR-Fusion* pipeline to align and map paired-end RNA-seq data from HGSC and normal tube samples. This suite requires several genomic resources that included: the reference genome (human genome version hg38), reference transcript structure annotations, and results from an all-vs-all BLAST+ search of reference transcript sequences [41]. 

Aligning RNA-Seq reads to the genome to capture split and discordant reads was done with *STAR* as an extension of the standard mapping procedure. First, the maximum mappable prefix algorithm was used to find the “seeds”, or read sequences, exactly matching to the genome. Then, genomic alignment windows were selected by clustering the anchor seeds. In each genomic window, a local alignment of the read sequence was performed. If the best alignment among all windows did not cover the entire read, chimeric detection was performed by finding the next best scoring window that covered the remainder of the read sequence. The *STAR-Fusion* mapping parameters are based upon the best practices for *STAR* [42], as well as parameters optimized to capture fusion transcripts, as described above [43]. Next, *STAR-Fusion* determines the most likely correct fusions, filtering out unlikely candidates from the initial predictions. More details can be found in Haas et al.’s publication [33]. Finally, we used *FusionInspector*, a component of the *STAR-Fusion* suite, that performs in silico validations of the fusion transcripts discoveries by performing a supervised analysis of fusion predictions [44] (Appendix B). 

### 4.6. Statistical Analysis

A table with all fusion transcripts and their annotation was constructed for all HGSC and tubal samples (Appendix A). Logistic regression was used to assess differences in fusion transcripts frequencies between HGSC and tubal samples. Fusion transcripts with statistical differences of *p* < 0.05 were introduced in a multivariate analysis to assess which fusion transcripts were independently significant between HGSC and tubal samples. To assess the association of survival with fusion transcripts, a survival analysis was performed using Cox proportional hazard ratios. A multivariate analysis of survival was built by introducing significant variables in the univariate analysis (*p* < 0.05) in a Cox Proportional Hazard ratio multivariate model. 

We additionally built a prediction model with fusion transcript data to determine which patients would have ovarian cancer (HGSC) versus those with normal fallopian tubes. To create this model, we used the lasso regression method, as implemented in the *glmnet* R package [45]. In our experience, lasso consistently lowers number of co-variates and computes area under the curve (AUC) with reliability and minimum errors, as compared to other prediction methods [46]. We evaluated the performance of our model using the AUC and its 95% confidence interval (CI). AUC was estimated with 1000 replicates of 10-fold cross-validation to avoid over-fitting of the model (internal validation) [47]. Bias-corrected and accelerated bootstrap CIs were computed for resulting AUCs. A value of 0.5 indicates a lack of model predictive performance, and 1.0 indicates perfect predictive performance, or the best model. For an alpha error of 0.05, a total sample size of 117 or more would be needed to create regression models of prediction with a power (1-Beta error) over 71% [48]. 

### 4.7. Validation of Fusion Transcript Detection with FusionCatcher

To validate the detection of fusion transcripts, we used another novel method, *FusionCatcher* [49,50], which has been also used to detect novel and known fusion transcripts in samples from patients with cancer. As above, data was aligned to the reference human genome version 38 (hg 38). Then, we performed C-statistics to determine the accuracy and AUC of our initial analysis for those fusion transcripts found to be significant in the. association analysis.

As a negative control for our analysis, we performed *STAR-Fusion* with *FusionInspector* validation in DNA from the normal tubal samples. We would expect not to find any fusion genes in normal tissue.

### 4.8. RT-PCR Validation of Fusion Transcripts

#### 4.8.1. RNA Purification

Whole cell RNA was purified from the appropriate tumors using the mirVana RNA purification kit according to manufacturer’s instructions (Thermo Fisher). Yield and purity were assessed in the University of Iowa Institute of Human Genetics (IIHG) using a Trinean DropSense 16 and an Agilent Model 2100 Bioanalyzer. 

#### 4.8.2. RT-PCR

The 500 ng of whole cell RNA from each tumor was reverse transcribed using the SuperScript III kit (Invitrogen). RT-PCR was performed on a BioRad T100 thermal cycler. Tumor cDNA was matched with the appropriate primers as shown below. * Tm is estimated in OligoAnalyzer (IDT) at 1.5 mM MgCl_2_.
Fusion TranscriptPCR Primer SequencesTm *CC2D1A-CPNE8For: ATGCACAAGAGGAAAGGACRev: GCAGGTGATGGCTTGATT59.3 °C59.7 °CFAM98B-FRMD5For: GTGCTGGACACACTGGAG Rev: TGCCGGGAAAGCAACAT61.5 °C61.6 °CAC004475.1-PRPF6For: GCAGCAGATGTACGACATGA Rev: CTTCAGGTTCTTCCAGCTCAA61.7 °C61.7 °CAUTS2-INO80CFor: CGGCAGAAGAGGACATCATTRev: CAGGTTCTTCCCAGGTTCTGTT 63.8 °C61.5 °C

Primers were designed using the appropriate fusion transcript sequence in PrimerQuest at Integrated DNA Technologies (idtdna.com). All primers were also manufactured by IDT. PCR amplifications were carried out for 35 cycles using an annealing temperature of 58.0 °C. In all cases, a negative control RNA was used consisting of whole cell RNA from a benign fallopian tube patient.

RT-PCR reactions were run on a 1.6% horizontal agarose gel. The gel was stained with ethidium bromide and visualized under uV irradiation on a Life Technologies E-Gel Imager (Thermo Fisher).

#### 4.8.3. Sequence Verification 

RT-PCR amplicons were purified using the QIAGEN QIAquick PCR Purification kit following manufacturer’s instructions (QIAGEN). Each amplicon was then sequenced on an Applied Biosystems Model 3730 × l capillary sequencer in the University of Iowa Institute of Human Genetics (IIHG) using the PCR primers as sequencing primers. Sequence output was then visualized in Finch TV software and validated by BLAST in ENSEMBL [21]. The RT-PCR amplicons were validated by direct Sanger sequencing.

## 5. Conclusions

In summary, we identified novel fusion transcripts that seem to be associated with HGSC and confer poorer survival to some of the patient with the disease. Further investigations will be needed to clarify the mechanisms of action of these transcripts and how they interact with survival. Some of these processes may be even targetable in future research.

## Figures and Tables

**Figure 1 ijms-22-04791-f001:**
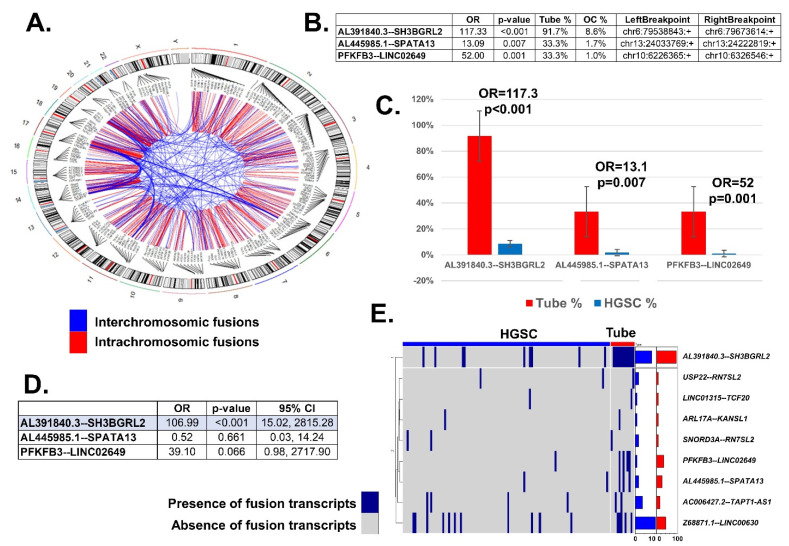
Fusion transcript differences between HGSC and normal fallopian tubes. (**A**). Circular chromosome representation of both components of the fused transcript; red: fusion of two transcripts within a chromosome (intrachromosomic); blue: fusion of two transcripts within a chromosome (interchromosomic). Due to space limitation, only unique fusion transcripts are depicted, and we omitted duplicated fusion transcripts. All names are not represented, but all connections are. (**B**) Fusion transcripts with different frequency between HGSC and tubal samples. Odds Ratio, *p*-value and location of the break are represented. (**C**) Frequencies of the 3 significant fusion transcripts in HGSC and tubal samples. The horizontal axis represents the frequency of these fusion transcripts in HGSC and tubal samples. Notably, these significant fusion transcripts are decreased in HGSC. Odds ratio, *p* value, and standard error for each transcript included. (**D**) Multivariate analysis comparing frequencies of fusion transcripts between HGSC and tubal samples: only *AL391840.3—SH3BGRL2* remained significant. (**E**) Heatmap representing HGSC and tubal samples (columns) with more differences in fusion transcript expression (*p* ≤ 0.1) in the analysis. In rows, presence (blue) or absence (grey) of fusion transcripts. The right of the panel contains two bar plots with the percentage of samples expressing these transcripts: blue represents HGSC samples, red represent normal tubes. Notice the scale of both bar plots: the percentage of these fusion transcripts found in HGSC samples (blue) is below 10%, while the percentage of these transcripts found in tubal samples (red) is over 15% more than half of the time.

**Figure 2 ijms-22-04791-f002:**
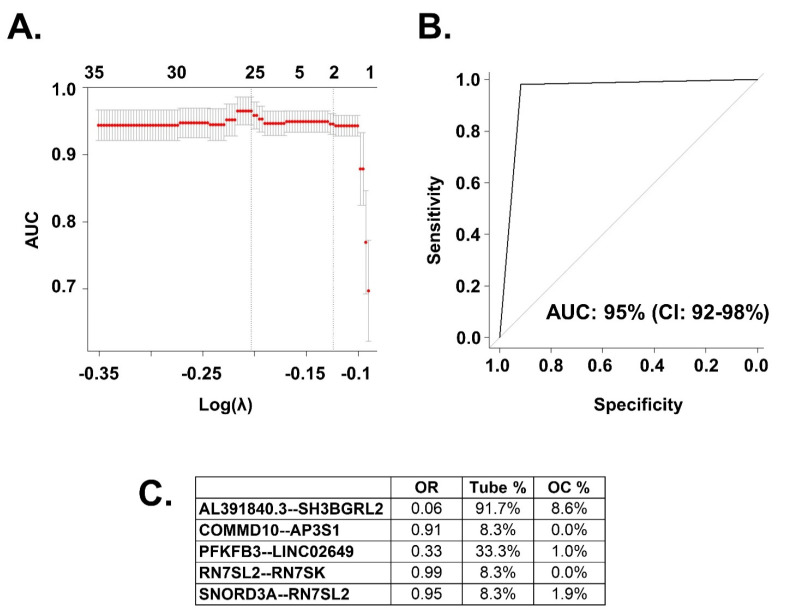
Prediction of HGSC with fusion transcripts from HGSC and tubal samples. (**A**) Model including all fusion transcripts. The best models are located in between the vertical dotted lines. Those will include between 2 and 25 fusion transcripts. The vertical axis represents the resulting AUC of the model with their 95% CI. The lower axis is the log of the λ, value used to optimize model construction. As detailed in methods, we performed 1000 bootstrap replicates to find the most adequate λ. (**B**) The best model with 5 fusion transcripts had an AUC of 95% with a 95% CI of 92%, 98%. (**C**) Fusion transcripts included in the model with their OR (HGSC vs. tube).

**Figure 3 ijms-22-04791-f003:**
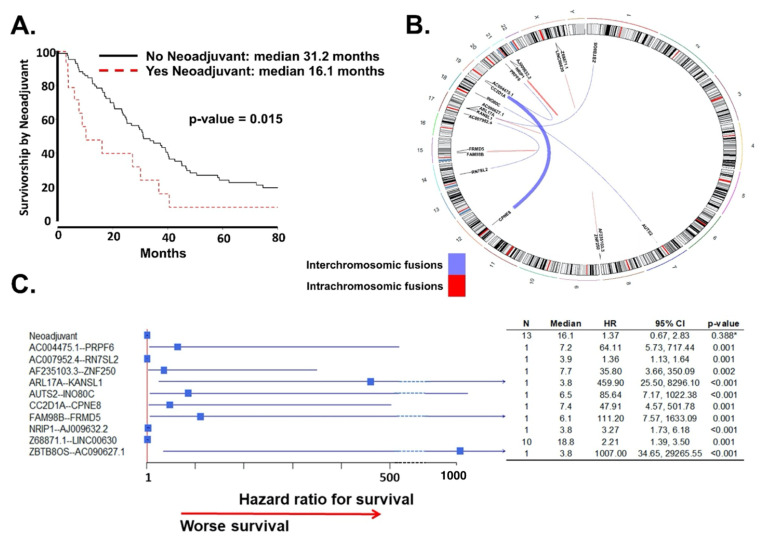
Association of fusion transcripts with survival. (**A**) In the multivariate analysis of survival for clinical variables, only neoadjuvant chemotherapy was significant. Representation of Kaplan–Meier survival curves between HGSC patients who received neoadjuvant chemotherapy vs. those that did not. *p*-value of the differences and median survivals are also shown (**B**) Circular chromosome representation of both components of the fused transcript; red: fusion of two transcripts within a chromosome (intrachromosomic); blue: fusion of two transcripts within a chromosome (interchromosomic). The width of the connecting line is in proportion of the number of reads observed for that particular fusion transcript: 37 reads observed in the patient with *CC2D1A*--*CPNE8* fusion and 15 for *NRIP1*--*AJ009632.2*. (**C**) Forest plot of the integrative multivariate model of survival that combined both independent significant clinical variables and fusion transcripts. All significant variables increased the risk of death by disease. N: number of HGSC patients with fusion transcripts; Median: median survival in months; HR: Hazard Ratio; CI: Confidence interval.

**Figure 4 ijms-22-04791-f004:**
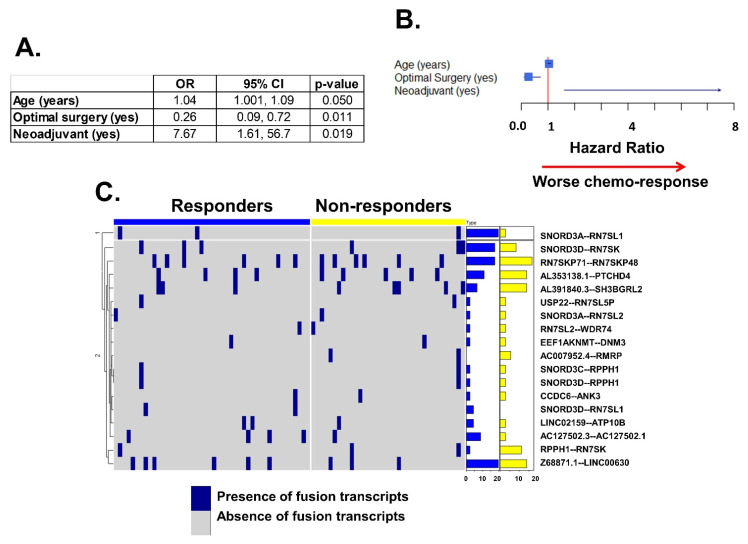
Association of fusion transcripts with chemo-response. (**A**) In the multivariate analysis, chemo-response was significantly associated with age, optimal surgery, and neoadjuvant chemotherapy. (**B**) Optimal cytoreduction is associated with better response to chemotherapy. The other two (age and neoadjuvant therapy) are associated with worse response to chemotherapy. (**C**) Heatmap representing responder and non-responder samples (columns) with more differences in fusion transcript expression (top 20) in the analysis. In rows, presence (blue) or absence (grey) of fusion transcripts. The right of the panel contains two bar plots with the percentage of samples expressing these transcripts: blue represents responder samples, yellow represent non-responder. The scale of both bar plots are similar. There is no difference of fusion transcripts between samples from responders vs. non-responder patients (chi-square *p*-value = 0.626).

**Figure 5 ijms-22-04791-f005:**
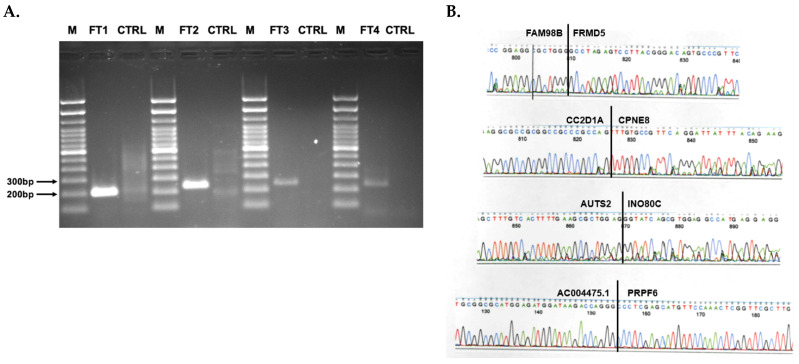
RT-PCR validation of fusion transcripts. (**A**) RT-PCR was performed on four fusion transcripts found to be significantly associated with overall survival in the multivariate analysis using *STAR-Fusion*/*FusionInspector* and *FusionCatcher*. FT1 represents *CC2D1A--CPNE8*, FT2 represents *FAM98B--FRMD5*, FT3 represents *AC004475.1--PRPF6*, and FT4 represents *AUTS2--INO80C.* Letter “M” represents the marker, which is the Invitrogen Track-it 100bp Ladder. (**B**) Chromatograms from direct sequencing of RT-PCR amplicons of four fusion transcripts shown in part A; FT2 is FAM98B--FRMD5, FT1 is CC2D1A--CPNE8, FT4 is AUTS2--INO80C, FT3 is AC004475.1--PRPF6.

**Table 1 ijms-22-04791-t001:** Validation of fusion transcript detection with *FusionCatcher*: These fusion transcripts found to be significant in the univariate analysis of survival were also observed with *FusionCatcher*. C statistics were performed to assess degree of concordance between both method, *STAR-Fusion* and *FusionCatcher*, for the 31 fusion transcripts detected with *FusionCatcher* alone. AUC: area under the curve; NPV: negative predictive value; PPV: positive predictive value.

*FusionCatcher*	AUC	Specificity	Accuracy	NPV	PPV
AP3D1--ARHGDIA	1	1	0.998	0.998	1
ARHGAP1--CKAP5	1	1	0.998	0.998	1
ARL17A--KANSL1	0.55	0.22	0.321	0.844	0.175
BTBD10--TEAD1	1	1	0.998	0.998	1
CC2D1A--CPNE8	1	1	0.998	0.998	1
CHTOP--PCAT1	1	1	0.998	0.998	1
DOT1L--GCGR	1	1	0.998	0.998	1
FAM20C--AC093627.4	1	1	0.998	0.998	1
FAM98B--FRMD5	1	1	0.998	0.998	1
FBXO34--SORCS3	1	1	0.998	0.998	1
GRIN2A--C16ORF72	1	1	0.998	0.998	1
INPP5B--PLEKHO1	1	1	0.998	0.998	1
LUC7L--AXIN1	1	1	0.998	0.998	1
MAGED2--ZFAT	1	1	0.998	0.998	1
MECOM--AC116337.3	1	1	0.998	0.998	1
NFE2L1--PNPO	1	1	0.998	0.998	1
NFKBIB--TEAD1	1	1	0.998	0.998	1
NRIP1--AJ009632.2	1	1	0.998	0.998	1
PACS1--HAUS3	1	1	0.998	0.998	1
PCAT1--C1ORF210	1	1	0.998	0.998	1
PGM2L1--POLD3	1	1	0.998	0.998	1
PSPC1--ZMYM5	0.50	0.20	0.231	0.948	0.052
RB1CC1--LINC02091	1	1	0.998	0.998	1
SMARCA4--ZNF700	1	1	0.998	0.998	1
TMCC1--CD96	1	1	0.998	0.998	1
TOGARAM1--FANCM	1	1	0.998	0.998	1
TRAPPC3--MAP7D1	1	1	0.998	0.998	1
TRMT1--CPA4	1	1	0.998	0.998	1
UBA2--RAD51B	1	1	0.998	0.998	1
UBE2F--LRRFIP1	1	1	0.998	0.998	1
ZNF609--SNX1	1	1	0.998	0.998	1

**Table 2 ijms-22-04791-t002:** Patient characteristics and association with survival: 103 patients with HGSC had complete clinical and outcome information for the survival analysis. * Statistically significant (later included in the multivariate analysis).

			HGSC Patients	HR	95% CI	*p*-Value
			N = 103			
Age	(mean)		59.8	1.01	0.99, 1.03	0.164
BMI	(mean)		27.2	1.00	0.97, 1.03	0.764
Preop CA-125	(mean)		2413.6	1.00	0.99, 1.00	0.488
Charlson Comorbidity Index	1–3		17	1.14	1.01, 1.31	0.044 *
	4–6		64			
	>6		18			
FIGO Stage	2		3	2.72	0, N/A	0.995
	3		68			
	4		25			
Disease in Upper abdomen (Other than Omentum) by Imaging	Yes	Large Bowel (N = 4)	63	1.60	1.02, 2.50	0.039 *
		Porta—Hepatis (N = 4)				
		Mesenteric Mets (N = 4)				
		Other (N = 26)				
	No		40			
Disease in the Chest by Imaging	Yes	Chest (N = 5)	7	1.11	0.44, 2.79	0.813
		Pleural effusion (N = 5)				
	No		96			
Grade	2		21	1.30	0.82, 2.07	0.270
	3		67			
Residual disease after surgery	Microscopic		20	0.59	0.32, 1.09	0.093
	Macroscopic		82			
	Optimal (<1 cm)		66	1.11	0.71, 1.73	0.639
	Suboptimal (>1 cm)		36			
Removal of Pelvic LN	Yes		17	1.83	0.27, 1.09	0.088
	No		86			
Removal of Para-Aortic LN	Yes		10	0.41	0.15, 1.11	0.080
	No		93			
Surgery of large bowel	Yes		29	1.43	0.91, 2.26	0.123
	No		74			
Surgical complexity score **	Low		52	1.58	0.56, 4.43	0.381
	Intermediate		47			
	High		4			
Neoadjuvant Chemotherapy	Yes		13	2.11	1.16, 3.83	0.015 *
	No		88			
Number of Cycles delivered	< 6		15	0.96	0.87, 1.07	0.476
	≥6		87			
Dose Dense Chemotherapy	Yes		3	0.60	0.15, 2.46	0.480

* Statistically significant (*p*-value < 0.05). ** Modification of Mayo complexity index: we did not have an entry of peritoneal or abdominal stripping; and rectosigmoidectomies with anastomosis were considered in our data collection as large bowel resections with anastomosis.

## Data Availability

Clinical data is not publicly available due to patient privacy. Datasets with RNA-seq can be browsed by their accession number: GSE156699. The validation part of this study was performed in silico, with de-identified publicly available data. All data from TCGA is available at their website: https://portal.gdc.cancer.gov/ (accessed on 1 January 2019). Software utilized by this study is also publicly available at Bioconductor website: http://bioconductor.org/ (accessed on 1 January 2019).

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
