# Peer review of "Identification of Novel Fusion Transcripts in High Grade Serous Ovarian Cancer"

_ijms, 2021, doi:10.3390/ijms22094791_

Round 1

Reviewer 1 Report

The manuscript is well written and

includes novel findings on fusion genes in

ovarian cancer. The standard deviations

in fig 1c can be included with statistics and p

value

Author Response

The manuscript is well written and includes novel findings on fusion genes in ovarian cancer. The standard deviations in fig 1c can be included with statistics and p value.

This has been updated. Thank you!

Reviewer 2 Report

The authors present their analysis of fusion genes in high grade serous ovarian cancer, and unaffected fallopian tubes and provide unanticipated suggestions that the presence of 3 fusion genes distinguish healthy viable fallopian tubes from ovarian cancer.  In another set of 10 fusion genes, the presence of fusion genes is associated with poor prognosis, which would follow the expected relationship of fusion genes in ovarian cancer.  The opposing "good" and "bad" fusion genes reported here would substantially benefit from confirmation of other fusion genes reported in previous manuscripts associated with poor prognosis with the current authors techniques.  For example, Mittal and McDonald (2015; 10.1186/s12920-015-0118-9) associate fusion genes with poor prognosis ovarian cancer. Cervera et al., 2021 identified fusion genes in ovarian cancer that affect the PI3K-AKT pathway and emergence of treatment resistance. CC2D1A is also described as Akt kinase interacting protein.  This adds a measure of confirmation credibility to CC2D1A fusion discovery that should be identified. NRIP1 modulates transcriptional activity of the estrogen receptor, which has known roles in fallopian tube. The discussion should dedicate a bit more effort disclosing potential connections with fallopian tube biology.  Paths to future research could suggest gene families regulated by partners of the fused genes to establish further validation of hyperactivate or hypoactive gene fusion products.

The emergence of multiple laboratory reports on fusion genes in high grade serous ovarian cancer specimens requires cross-laboratory confirmation. Further validation of the current fusion gene findings that confirms a subset of previous fusion gene discoveries with the current RT-PCR approach would substantially improve this report and integrate recently published results on this same topic. 

It would be nice to see one of the authors selected fusion genes to be highlighted in a version of Figure 1A in Supplemental Results for readers to be able to follow the fusion genes associated with normal fallopian tube.  Further, it would be useful to understand if the fused portion of genes in normal fallopian tube is limited in size relative to fusion products associated with ovarian cancer.  In other words, is their a fusion size that distinguishes cancer tissues. 

Author Response

Reviewer 2

The authors present their analysis of fusion genes in high grade serous ovarian cancer, and unaffected fallopian tubes and provide unanticipated suggestions that the presence of 3 fusion genes distinguish healthy viable fallopian tubes from ovarian cancer.  In another set of 10 fusion genes, the presence of fusion genes is associated with poor prognosis, which would follow the expected relationship of fusion genes in ovarian cancer.  The opposing "good" and "bad" fusion genes reported here would substantially benefit from confirmation of other fusion genes reported in previous manuscripts associated with poor prognosis with the current authors techniques.  For example, Mittal and McDonald (2015; 10.1186/s12920-015-0118-9) associate fusion genes with poor prognosis ovarian cancer. Cervera et al., 2021 identified fusion genes in ovarian cancer that affect the PI3K-AKT pathway and emergence of treatment resistance. CC2D1A is also described as Akt kinase interacting protein.  This adds a measure of confirmation credibility to CC2D1A fusion discovery that should be identified. NRIP1 modulates transcriptional activity of the estrogen receptor, which has known roles in fallopian tube. The discussion should dedicate a bit more effort disclosing potential connections with fallopian tube biology.  Paths to future research could suggest gene families regulated by partners of the fused genes to establish further validation of hyperactivate or hypoactive gene fusion products.

We agree that identifying common fusion transcripts also found in other publications would add validity to this work. Indeed, we re-queried our ovarian cancer fusion transcript database, and we found some fusion transcript partners in common with the referenced papers, though not the fusion transcripts themselves. From the transcripts highlighted in Mittal and McDonald, 2005, DNAJB1 was found to be a fusion partner in our dataset, but its partners were ENSG00000148719.15 and POLR3A, not MED26. MED26 partnered with ENPP2 only. FARP1 partnered with MBNL2, not SLC15A1. None of these transcripts were significantly associated with poor prognosis. From the Cervera et al 2021 paper, AKT2 was found to partner with DNAH17 in our dataset. PTK2 was found to partner with TRAPPC9.

We agree that discussing NRIP1’s involvement with transcriptional activity of the estrogen receptor  and fallopian tube development specifically is a salient point to this paper, and we have included this in the discussion.

The emergence of multiple laboratory reports on fusion genes in high grade serous ovarian cancer specimens requires cross-laboratory confirmation. Further validation of the current fusion gene findings that confirms a subset of previous fusion gene discoveries with the current RT-PCR approach would substantially improve this report and integrate recently published results on this same topic. 

We absolutely agree with this point. However, given time and resource limitations, we performed the RT-PCR validation on a subset of transcripts.

It would be nice to see one of the authors selected fusion genes to be highlighted in a version of Figure 1A in Supplemental Results for readers to be able to follow the fusion genes associated with normal fallopian tube. 

We have included an additional figure.

Further, it would be useful to understand if the fused portion of genes in normal fallopian tube is limited in size relative to fusion products associated with ovarian cancer.  In other words, is their a fusion size that distinguishes cancer tissues. 

Thank you for this suggestion. We have completed this analysis and found that the fusion transcripts in the normal fallopian tubes were significantly shorter than those found in the ovarian cancer. We have included this analysis as Supplemental Table S5.